# PromptARA: Improving Deep Representation in Hybrid Automatic Readability Assessment with Prompt and Orthogonal Projection

**Jinshan Zeng[1], Xianglong Yu[2], Xianchao Tong[2], Wenyan Xiao[3,*]**

[1]School of Computer and Information Science, Jiangxi Normal University
[2]School of Digital Industry, Jiangxi Normal University
[3]School of Foreign Languages, JiangXi University of Science and Technology
`jinshanzeng@jxnu.edu.cn, xianglongyu@jxnu.edu.cn,`
`xianchaotong@jxnu.edu.cn, wy.xiao@jxust.edu.cn`

## Abstract

Readability assessment aims to automatically classify texts based on readers' reading levels. The hybrid automatic readability assessment (ARA) models using both deep and linguistic features have attracted rising attention in recent years due to their impressive performance. However, deep features are not fully explored due to the scarcity of training data, and the fusion of deep and linguistic features is not very effective in existing hybrid ARA models. In this paper, we propose a novel hybrid ARA model called **PromptARA** through employing prompts to improve deep feature representations and an orthogonal projection layer to fuse both deep and linguistic features. A series of experiments are conducted over four English and two Chinese corpora to show the effectiveness of the proposed model. Experimental results demonstrate that the proposed model is superior to state-of-the-art models.

## 1  Introduction

Text readability assessment aims to quantify the difficulty of a text, that is, the degree to which it can be easily read and understood (McLaughlin, 1969; Klare, 2000). Due to the superiority of an automatic readability assessment (ARA) system on assigning a text to a difficulty grade, ARA is useful for identifying texts or books that are suitable for individuals according to their language proficiency, intellectual and psychological development. The studies on ARA can be traced back to the last century (Lively and Pressey, 1923; Klare, 1963) and have attracted rising attention in recent years, with impressive performance yielded by many neural approaches (Tseng et al., 2019; Schicchi et al., 2020; Azpiazu and Pera, 2019; Deutsch et al., 2020; Martinc et al., 2021; Lee et al., 2021; Vajjala, 2022; Tanaka-Ishii et al., 2010; Lee and Vajjala, 2022; Zeng et al., 2022).

In the early stage, studies in ARA mainly focused on readability formulas, which are typically developed through empirical pedagogy and psychology (Klare, 1963; Davison and Kantor, 1982). Although these formulas are easily interpretable, they rely on surface features and cannot measure the structure or semantic complexity of a text, resulting in unsatisfactory performance.

Then, the traditional machine learning methods have been applied to train statistical classifiers for ARA. These classifiers exploit a large number of features at various levels of a text, including but not limited to vocabulary, semantics and syntax (Hancke et al., 2012; Sung et al., 2015; Dell'Orletta et al., 2011; Francois and Fairon, 2012; Denning et al., 2016; Arfé et al., 2018; Jiang et al., 2019). Although they often achieve better performance than readability formulas, the feature engineering and selection for these machine learning methods are generally time-consuming and labor-intensive.

In recent years, deep learning methods have shown impressive performance in natural language processing and their application in ARA has been intensively studied. Pre-trained word embedding models (Mikolov et al., 2013; Pennington et al., 2014; Bojanowski et al., 2017) and masked language models such as BERT (Devlin et al., 2019) have been exploited by many neural ARA models (Deutsch et al., 2020; Tseng et al., 2019; Zeng et al., 2022). However, the performance of these deep learning methods is limited by the scarcity of training data in many ARA tasks.

To further improve the performance of neural approaches based on deep learning, the hybrid ARA models utilizing both linguistic and deep features have been recently studied in the literature (Qiu et al., 2018; Deutsch et al., 2020; Lee et al., 2021; Li et al., 2022). The literature (Deutsch et al., 2020; Lee et al., 2021) investigated the joint effect of handcrafted linguistic features and deep features extracted by a deep neural network, where deep features and hand-crafted linguistic features were simultaneously fed into a machine learning model

without fusion to yield readability levels. The recent literature (Li et al., 2022) introduced certain difficulty-aware topic features through utilizing the word difficulty knowledge to guide the training of topic model, and fused the deep features and linguistic features by a projection scheme.

Although the hybrid ARA models have achieved the state-of-the-art performance, there are two limitations for these models. The first one is that deep features are not fully explored in hybrid ARA models due to the scarcity of training data. In most of existing neural ARA models such as Zeng et al. (2022), deep features are generally extracted by the pre-trained large language models, which commonly require a large amount of training data. The second one is that the fusion of deep features and linguistic features in existing hybrid ARA models is not paid particular attention.

In this paper, we employ prompts to improve deep feature representations inspired by the great success of prompt learning (Lee and Lee, 2023; Liu et al., 2023; Schick and Schütze, 2021), and an orthogonal projection layer to effectively fuse the improved deep features and linguistic features and particularly remove the redundant information among these features. The major contributions of this paper can be summarized as follows.

- We propose a novel hybrid ARA model called **PromptARA** through exploiting prompts to improve the extraction of deep features, and an orthogonal projection layer to effectively fuse linguistic features and deep features at various levels of a text, including word, sentence and document levels. The proposed model can focus more on some in-domain information of the text with the help of prompts and reduce the redundant information among different levels of feature representations.

- Extensive experiments are conducted over four English benchmark corpora and two Chinese corpora to validate the merits of the proposed model through comparing with many state-of-the-art models. Experimental results demonstrate that the employed prompts and orthogonal projection layers are effective and that the proposed model is superior to state-of-the-art models in most of corpora in terms of several important evaluation metrics.

## 2 Related Work

Text exhibits an inherent structure, and its difficulty is manifested through various linguistic levels, encompassing words, sentences, and entire documents. How to incorporate more effective information of text into ARA is one of the current research directions of ARA. As the current mainstream models, deep ARA models is mainly based on the hierarchical attention networks (HAN) and pre-trained models (Yang et al., 2016; Azpiazu and Pera, 2019; Zeng et al., 2022). The basic idea of such kind of models is to use multi-level deep representations of texts to yield better representations for prediction. In Azpiazu and Pera (2019), a hierarchical attention model called Vec2Read was suggested for the multilingual ARA task, where the word-level information such as the grammatical and morphological information was introduced. In the recent literature (Zeng et al., 2022), a novel HAN-type model was suggested mainly based on the idea of ordinal regression.

Noticing that deep features extracted by the kind of HAN models generally have poor interpretability and their capacity is limited by the scarcity of training data, the hybrid ARA models by using both linguistic features and deep features have attracted rising attention in recent years (Deutsch et al., 2020; Qiu et al., 2021; Lee et al., 2021; Li et al., 2022). In Deutsch et al. (2020), the authors directly used deep and linguistic features to jointly train a statistical classifier. In Qiu et al. (2021), the authors firstly yielded an improved linguistic representation by learning the relevance of the used linguistic features and then fused the improved linguistic representation with the deep representation by a simple concatenation to yield a new representation for prediction. The recent literature (Lee et al., 2021) deeply investigated the effectiveness of the linguistic features when incorporated into the deep models for ARA. In Li et al. (2022), the authors firstly introduced a deep topic feature through introducing the difficulty knowledge, and then suggested a projection scheme to fuse the deep and linguistic features by removing redundant information.

Although existing hybrid ARA models achieve the sate-of-the-art performance, the deep features used in these models are extracted by some pre-trained language models, which generally require a large amount of training data when adapted to downstream tasks. Thus, the performance of these

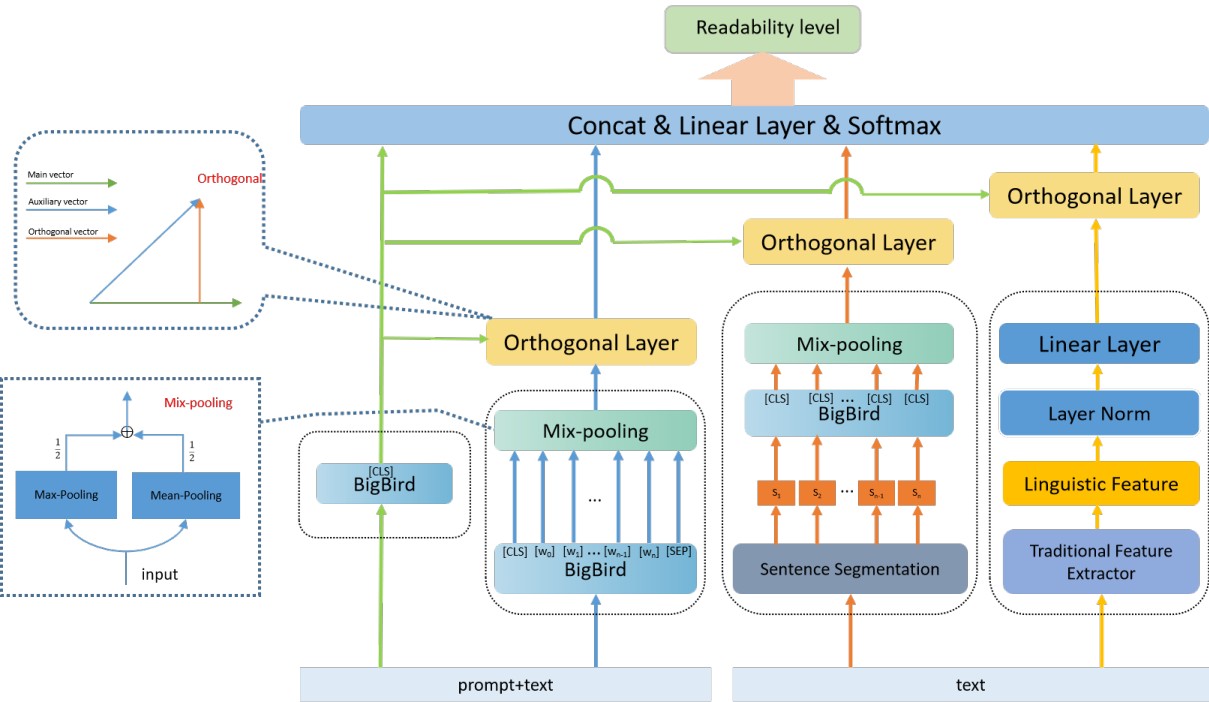

Figure 1: Overview of the proposed model.

hybrid ARA models is limited by the scarcity of training data. The recently suggested prompt learning (Schick and Schütze, 2021) provides a potential way to address this issue. In the recent literature (Lee and Lee, 2023), the authors introduced prompt learning to the problem of judging the difficulty for two given texts, which can be roughly regarded as a binary classification problem, while the ARA task considered in this paper can be generally regarded as a multi-classification task. Inspired by the great success of prompt learning, this paper employs prompts to improve the deep representations represented in multiple levels, and an orthogonal projection scheme to effectively fuse the deep and linguistic features for prediction.

## 3 Proposed Model

In this section, we describe the proposed model in detail. The overview of the proposed model is depicted in Figure 1. As shown in Figure 1, four types of features including the document-, sentence- and word-level deep features and the traditional linguistic features are utilized for prediction. The document- and word-level deep features are extracted by texts with prompts, while the sentence-level deep features and traditional linguistic features are extracted from the original texts. The prompt guided document-level features are utilized to improve the other three types of features via an

orthogonal projection layer. With these refined four types of features, the prediction of the readability level for a given text is yielded by a linear classifier.

### 3.1 Deep Representations with Prompts

To fully explore deep features from texts, we consider three levels of deep features, where the document- and word-level deep features are extracted from the extended texts with prompts while the sentence-level deep features are yielded from original texts. For the sentence-level deep features, we extract them directly from original texts since the prompts designed in this paper are a few sentences, which may provide little help for the extraction of sentence-level deep features. Some prompts are presented in Table 1.

**A. Document-level deep representations.** As depicted in Figure 1, we employ the well-known pre-trained Transformers for Longer Sequences (BigBird) model (Zaheer et al., 2020) to extract the document-level deep features $f_d$ from extended texts with prompts, inspired by the advantage of BigBird model on dealing with long sequences.

**B. Word-level deep representations.** To yield the word-level deep representations, we firstly feed extended texts with prompts to a BigBird to obtain word-level embeddings and then implement a mix-pooling operation for word-level embeddings. As depicted in Figure 1, the mix-pooling opera-

| Prompts |
|---|
| The text readability classification task is currently in progress. |
| The task is to divide the text into difficulty levels. |
| After reading this article, give the difficulty level of the article. |
| The readability task is currently underway to determine the difficulty of the text. |

Table 1: Some examples of prompts designed for ARA. This paper uses the first prompt as the default one.

tion is the mixing of the well-known max-pooling and mean-pooling operations, aiming to extract the maximum and mean values.

**C. Sentence-level deep representations.** To yield the sentence-level deep representations, we firstly segment original texts into single sentences, then feed them into BigBird to yield sentence-level embeddings, and later implement the mix-pooling for sentence-level embeddings.

### 3.2 Linguistic Feature Representations

Many previous studies have shown that linguistic features can provide additional textual information for the deep neural models and thus improve their performance (Sennrich and Haddow, 2016; Qiu et al., 2018; Lee et al., 2021; Li et al., 2022).

As depicted in Figure 1, to yield the linguistic feature representations, we firstly extract a large number of linguistic features at various levels by the traditional feature extractors, where the specific linguistic features for the Chinese texts are presented in Tables 8 and 9 in Appendix. For English traditional features, we extract them by implementing the lingfeat toolkit developed in Lee et al. (2021) . Then, we implement the layer normalization for linguistic features for the purpose of training stability. Finally, we yield the linguistic representations by projecting the normalized linguistic features into a common space with the same dimension of deep representations.

### 3.3 Fusion by Orthogonal Layers

Noticing that there is redundant information between deep and linguistic representations, we employ the orthogonal projection layer to remove the redundant information, inspired by the literature (Li et al., 2022). As depicted in Figure 1, given a main representation and an auxiliary representation, we implement orthogonal projection for these two representations, that is, we keep the orthogonal part of the auxiliary representation while eliminating the projection part of the auxiliary representation along the main representation, with the purpose of

getting rid of the redundant information between these two different levels of representations. In this paper, we take the document-level representation as the main representation and the other three levels of representations as the auxiliary representations, and implement the orthogonal projection operation for the other three levels of representations to keep their orthogonal parts as important supplemental information for the main representation.

With these representations after orthogonal layers, we concatenate them and feed them into a linear layer followed by the softmax as a statistical classifier to yield the prediction of readability level.

## 4 Experimental Settings

In this section, we describe the experimental settings in detail. We firstly describe the six corpora used in the experiments including four English benchmark corpora and two Chinese corpora, and then present the baseline models, and finally describe some implementation details.

### 4.1 Corpora

We conducted experiments on four English corpora and two Chinese corpora to demonstrate the effectiveness of the proposed model. Some statistics of these corpora are presented in Table 2.

**Weebit** (Vajjala and Meurers, 2012). The Weebit corpus is a combined five-level corpus created based on WeeklyReader[1] and BBC-Bitesize[2], containing a total of 6388 texts. It is often considered as the gold standard for ARA models. For each difficulty category we perform 625 text downsampling.

**Cambridge**[3] (Xia et al., 2016). The levels of the Cambridge English tests (KET, PET, FCE, CAE, CPE) are used to categorize articles. For each difficulty category, we performed downsampling on 60 texts.

---

[1]http://www.weeklyreader.com
[2]http://www.bbc.co.uk/bitesize
[3]http://www.cambridgeenglish.org

| Properties | Weebit | Cambridge | Newsela | CLEAR | CMER | CMT |
|---|---|---|---|---|---|---|
| Language category | English | English | English | English | Chinese | Chinese |
| Number of classes | 5 | 5 | 11 | 10 | 12 | 12 |
| Number of texts | 3125 | 300 | 9565 | 4724 | 2260 | 2621 |
| Average length | 288 | 510 | 747.37 | 171.96 | 926.94 | 674.72 |

Table 2: Statistics for the used English and Chinese corpora.

**Newsela**[4] (Xu et al., 2015). Newsela contains 10,786 texts, out of which we selected 9565 texts in English. The dataset is a parallel corpus of original and simplified document alignment versions, corresponding to 11 different unbalanced grade levels (from grade 2 to grade 12).

**CLEAR**[5] (Crossley et al., 2022). The corpus comprises 4724 text excerpts and offers a distinct measure of readability for each text, designed for readers in grades 3-12. It also includes metadata such as the year of publication, genre, and other relevant information for the excerpts. The CLEAR corpus represents a significant advancement over previous readability corpora in terms of the size and diversity of available excerpts. It encompasses more than 250 years of writing across two different genres, and provides a unique criterion for readability based on teachers' evaluations of text difficulty for student readers. To adapt to the task at hand, we utilized the Lexile Band, resulting in the creation of ten classes.

**CMT** (Cheng et al., 2020). The CMT consists of texts from Chinese textbooks used in mainland China, ranging from the first grade of primary school to the third grade of high school. This corpus comprises a total of 2,621 texts.

**CMER**[6] (Zeng et al., 2022). CMER was collected by Zeng et al. (2022) from extracurricular books targeted towards children and young adult in the mainland Chinese book markets. The corpus consists of 2,260 texts, which are categorized into 12 categories, aligning with grades 1 through 12.

### 4.2 Baselines

We consider the following state-of-the-art models as the baselines to verify the effectiveness of the proposed model.

**BERT**[7] [8] (Devlin et al., 2019) represents fine-tuning using the default BERT model.

**HAN** is a model based on hierarchical attention

networks proposed by Yang et al. (2016). It employs static word embeddings and utilizes two hierarchical attention mechanisms at both the word and sentence levels to pay attention to salient words and sentences. This enables the model to assign varying levels of attention to content of different importance when constructing text representations. We adopted the same framework used by Martinc et al. (2021), who replaced Bi-GRU with Bi-LSTM.

**BigBird**[9] [10] (Zaheer et al., 2020) is a model based on Transformer with sparse attention mechanism, which can handle up to 4096 tokens and get better performance. We use it for ARA for fine-tuning.

**DTRA** (Zeng et al., 2022) is a model based on BERT word embeddings with a HAN-like structure. It learns the sequential information of inter-textual difficulty by predicting the relative difficulty of paired texts and using distance-dependent soft labels.

**Lee-2021** (Lee et al., 2021) is a hybrid ARA model, where three novel features are proposed in terms of high-level semantics, and the deep and linguistic features are directly used to train a statistical classifier.

**BERT-FP-LBL** (Li et al., 2022) is a hybrid ARA model that utilizes the orthogonal projection to fuse the linguistic features and deep features.

### 4.3 Implementation Details

For two Chinese corpora, we extracted 67 linguistic features at different levels including lexical, semantic, syntactic, and cohesion aspects. As for the four benchmark English corpora, we extracted the linguistic features using the toolkit suggested in Lee et al. (2021) at various levels such as the discourse, syntactic, lexical and surface levels.

We evaluated the proposed model in terms of the following four commonly used metrics for classification, i.e., *accuracy* (Acc), *precision* (Pre), the *macro F1-metric* (F1), *Quadratic Weighted Kappa* (QWK), where the QWK metric is often used to

---

[4] https://newsela.com
[5] https://github.com/scrosseye/CLEAR-Corpus
[6] https://github.com/JinshanZeng/DTRA-Readability
[7] https://huggingface.co/bert-base-uncased
[8] https://huggingface.co/bert-base-chinese

[9] https://huggingface.co/google/bigbird-roberta-base
[10] https://huggingface.co/Lowin/chinese-bigbird-base-4096

| Dataset | Cambridge | Weebit | CLEAR | Newsela | CMER | CMT |
|---|---|---|---|---|---|---|
| MaxLen | 1536 | 1536 | 512 | 2048 | 2048 | 2048 |
| Epoch | 30 | 30 | 30 | 30 | 20 | 20 |
| Learn. rate | 1e-5 | 1e-5 | 1e-5 | 1e-5 | 2e-5 | 2e-5 |

Table 3: Hyperparameter settings for PromptARA.

assess consistency or reliability among multiple assessors, and it provides more accurate insights and decisions in the area of readability assessment.

We used the AdamW optimizer (Loshchilov and Hutter, 2017) with a weighting decay parameter 0.01 and a warm-up ratio 0.1 to train the proposed model. All experiments were conducted on RTX 3090 and A40 GPUs, and implemented using the PyTorch framework. Some hyperparameters of the proposed model are presented in Table 3.

In particular, since we cannot access the reproducible source codes, it is not easy to reproduce the results in the recent literature in Lee et al. (2021) and Li et al. (2022). Thus, we directly took their reported results as the comparison results. For fair comparisons, we followed the similar settings of Lee et al. (2021) and Li et al. (2022) to implement other baselines. Specifically, for each corpus, we divided it into the training, validation and testing sets using an 8:1:1 ratio for three times. The experimental results were recorded on average by running three trails.

## 5 Experimental Results

In this section, we describe and analyze the experimental results in detail.

### 5.1 Performance Evaluation

Tables 4 and 5 present the experimental results of the proposed model and baselines on the English and Chinese datasets, respectively. Since these two Chinese corpora are not considered in the literature Lee et al. (2021) and Li et al. (2022), we do not compare these two baselines over the Chinese corpus as shown in Table 5.

**A. Evaluation over English Corpora.** When regarding the performance over English corpora, we can observe that the proposed model outperforms these baselines in most of cases. Specifically, when compared to the base model BigBird using only the document-level representations from original texts, the proposed model yields much improvement over all English corpora. This demonstrates that the suggested prompts, word-level and sentence-level deep representations, as well the linguistic features in the proposed model are useful for ARA.

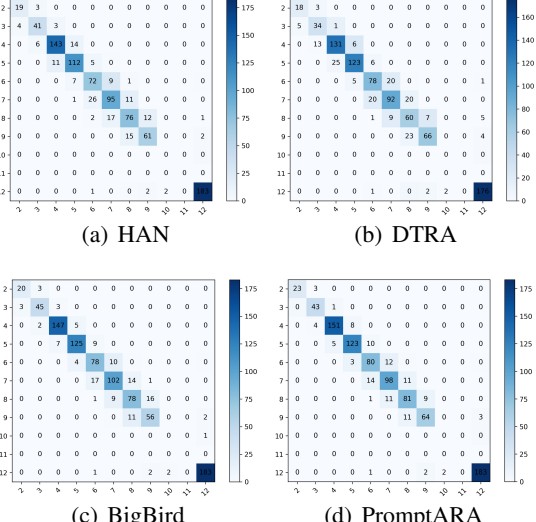

|   |   |   |   |
|---|---|---|---|
| (a) HAN | | (b) DTRA | |
| (c) BigBird | | (d) PromptARA | |

Figure 2: Visualization results of confusion matrices of four deep learning models over Newsela, where the horizontal- and vertical-axis of each figure represents the predicted categories and the true categories of the samples, respectively.

When compared to these two HAN-type models, i.e., HAN and DTRA, the proposed model significantly outperforms these two deep models with HAN-like structures. It is noteworthy that the multi-level deep representations in HAN and DTRA are represented and fused in a hierarchical way, whereas the multi-level deep representations in the proposed model are yielded in a parallel way and fused by an orthogonal projection scheme, together with the linguistic features. These results show that the proposed model can explore more deep features for ARA.

Compared to these two hybrid ARA models of Lee et al. (2021) and Li et al. (2022) using both deep and linguistic features, the proposed model also outperforms them over the Cambridge and Weebit corpora in terms of most evaluation metrics. We can observe from Table 4 that except the QWK value on Weebit, the proposed model achieves the best results for all other cases. This shows the effectiveness of the proposed models in comparison of existing hybrid ARA models.

Moreover, it can be observed from Table 4 that the performance of BigBird is much better than BERT, another pre-trained model used in ARA, mainly due to the superiority of BigBird on dealing with longer sentences.

We further provide some visualization results in terms of confusion matrices to demonstrate the ef-

| Model | | HAN | BERT | DTRA | BigBird | Lee2021* | BERT-FP_LBL* | PromptARA |
|---|---|---|---|---|---|---|---|---|
| Cambridge | Acc | 76.67 | 75.56 | 77.78 | 87.78 | 76.30 | 87.78 | **91.11** |
| | Pre | 80.49 | 72.75 | 79.71 | 88.29 | 79.20 | 89.46 | **92.24** |
| | F1 | 75.77 | 72.95 | 77.07 | 87.53 | 75.20 | 87.73 | **90.88** |
| | QWK | 92.64 | 91.59 | 92.62 | 97.04 | 91.90 | 96.87 | **97.82** |
| weebit | Acc | 82.54 | 91.53 | 85.29 | 92.70 | 90.50 | 92.70 | **93.12** |
| | Pre | 83.73 | 91.56 | 85.54 | 92.73 | 90.50 | 92.89 | **93.19** |
| | F1 | 82.76 | 91.51 | 85.30 | 92.70 | 90.50 | 92.73 | **93.09** |
| | QWK | 94.48 | 97.10 | 95.65 | 97.17 | 96.80 | **97.78** | 97.43 |
| CLEAR | Acc | 67.87 | 76.74 | 72.09 | 78.86 | - | - | **82.03** |
| | Pre | 66.22 | 76.23 | 70.75 | 79.01 | - | - | **81.99** |
| | F1 | 66.43 | 76.05 | 70.81 | 78.34 | - | - | **81.75** |
| | QWK | 89.29 | 92.86 | 91.85 | 94.03 | - | - | **94.54** |
| Newsela | Acc | 83.80 | 77.12 | 83.07 | 87.15 | - | - | **88.40** |
| | Pre | 83.86 | 78.45 | 82.96 | 87.14 | - | - | **88.21** |
| | F1 | 83.70 | 76.59 | 82.82 | 87.05 | - | - | **88.24** |
| | QWK | 98.35 | 97.67 | 98.41 | 98.79 | - | - | **98.88** |

Table 4: Comparison results of the proposed PromptARA model and baselines over four English benchmark corpora. * Experimental results taken directly from the literature. The best results are marked in bold.

| Model | | HAN | BERT | DTRA | BigBird | PromptARA |
|---|---|---|---|---|---|---|
| CMT | Acc | 42.53 | 38.46 | **44.42** | 38.46 | 43.96 |
| | Pre | 40.57 | 38.79 | **44.24** | 38.89 | 43.17 |
| | F1 | 41.09 | 37.17 | **43.87** | 37.81 | 41.60 |
| | QWK | 88.00 | 88.09 | 89.95 | 89.52 | **91.20** |
| CMER | Acc | 23.40 | 22.30 | **26.50** | 23.84 | **26.50** |
| | Pre | 15.47 | 30.13 | **25.36** | 24.70 | 24.24 |
| | F1 | 18.48 | 13.49 | **25.16** | 23.57 | 23.92 |
| | QWK | 72.10 | 65.39 | 70.53 | **73.74** | 68.74 |

Table 5: ARA performance on the Chinese datasets. The best and second best results are marked in bold and blue color, respectively.

fectiveness of the proposed model over Newsela, as depicted in Figure 2. We can observe from Figure 2 that the confusion matrix yielded by the proposed model is more concentrated to the diagonal. This also shows the effectiveness of the proposed model as compared to the baselines.

**B. Evaluation over Chinese Corpora.** When evaluating the performance of the proposed model over these two Chinese corpora, i.e., CMT and CMER, we can observe that the proposed model achieves the competitive results in comparison of baselines. Specifically, the proposed model achieves the best result in terms of the QWK and the second best results in terms of the other three evaluation metrics over CMT corpus, while yields the best classification accuracy over CMER.

The performance behaviour of the proposed model might be attributed to the following two main factors. Firstly, the Chinese variant of the Big-Bird model was not well trained on a large amount of data, limiting its ability to adapt to the complex and diverse nature of Chinese texts. Secondly, the complexity of Chinese texts is influenced by intricate semantic and structural differences, further

contributing to the observed performance difference.

## 5.2 Ablation Studies

In this subsection, we conducted a series of ablation studies over two English datasets (Cambridge and CLEAR) and two Chinese datasets (CMT and CMER) to validate the effectiveness and feasibility of our proposed ideas. In view of the deficiencies in the Chinese dataset in the comparative experiments, we also used other pre-trained models to carry out relevant experiments. Besides the proposed PromptARA model, we considered the following three models:

• **w/o L** denotes the model that does not use the linguistic features in PromptARA.

• **w/o (L, S, W)** denotes the model that does not use the linguistic features, the sentence-level and word-level deep representations in PromptARA.

• **w/o (L, S, W, P)** denotes the model that does not use the linguistic features, the sentence-level and word-level deep representations as well as prompts in PromptARA.

| Model | CLEAR | Cambridge | CMER | CMT |
|---|---|---|---|---|
| PromptARA | **82.03** | **91.11** | 26.50 | **43.96** |
| w/o L | 81.61 | 91.11 | **26.71** | 41.39 |
| w/o (L, S, W) | 80.97 | 88.89 | 26.05 | 39.01 |
| w/o (L, S, W, P) | 78.86 | 87.78 | 25.61 | 35.16 |

Table 6: Experimental results for ablation studies over four corpora in terms of accuracy, where **L**, **S**, **W** and **P** represent respectively the linguistic features, sentence-level representations, word-level representations and prompts used in the proposed model. The best results are marked in bold.

The experimental results are presented in Table 6. From Table 6, the accuracy gradually decays over most corpora through removing the linguistic features, sentence- and word-level features, and prompts from the proposed PromptARA model in sequence. In particular, the introduced prompt information generally yields the biggest improvements on accuracy as shown in the fourth and fifth rows in Table 6. As shown in the third and fourth rows in Table 6, the introduced word- and sentence-level representations are useful for ARA. In addition, we can observe from the second and third rows of Table 6 that linguistic features are also useful to refine the deep features in most cases. These results clearly verify the effectiveness and feasibility of the proposed ideas.

Moreover, we consider the effect of the language models in the proposed model. Specifically, we compare the performance of the proposed model with its counterparts by replacing the Big-Bird model with other pre-trained models such as CINO[11] (Yang et al., 2022), BERT[12], Muiti-BERT [13], Longformer [14] (Iz et al., 2020), over these two Chinese corpora. The comparison results are presented in Table 7. As shown in Table 7, the proposed model integrated with the BigBird model achieves the best accuracy, attributed to its superiority on handling long length texts and outperforms the Longformer model, which also has the ability to handle long texts. For the most other pre-trained language models, they generally have to truncate the texts during the input procedure, resulting in a greater loss of valuable textual information and thus the degradation of the performance.

| Dataset | BERT | CINO | Multi-BERT | Longformer | BigBird |
|---------|------|------|------------|------------|---------|
| CMT | 41.23 | 38.10 | 38.64 | 42.86 | **43.96** |
| CMET | 24.14 | 25.13 | 25.44 | 23.62 | **26.50** |

Table 7: On effect of pre-trained language models used in the proposed model. The best results are marked in bold.

## 6 Conclusion

How to fully explore the deep representations and exploit the linguistic representations of texts is important for ARA. This paper proposed a novel hybrid ARA model through employing prompts to improve deep representations and fusing the multi-level deep and linguistic representations with an orthogonal projection scheme. The proposed model uses informative prompts to provide additional information to enhance the contextual semantic understanding of the model, and combines textual information about the different structures of the text to yield a richer textual representation. Experimental results show that the proposed model achieves excellent performance on both Chinese and English datasets compared with existing approaches. On the English datasets in particular, it shows better generalization performance and robustness. In future work, we will further explore how to achieve automatic generation of textual prompts. Capturing richer and more accurate information representation at different text levels will also be investigated.

## Limitations

The model poses a major challenge due to the need to obtain vectors with different structural levels, which is more demanding on the equipment. Therefore, it is crucial to explore a model that runs on low-demanding devices without compromising the performance of the model. This becomes particularly important when dealing with large amounts of distributed text and large datasets, which must have higher equipment requirements. In addition, we are currently having difficulty deciding which prompt texts are appropriate. Another challenge we face is how to adapt the model to feature selection based on the characteristics of the dataset itself. Different datasets exhibit unique features, where too many features introduce too much noise and too few features do not have much impact on the performance of the model.

## Acknowledgements

The work of J. Zeng was supported in part by the National Natural Science Foundation of China [Grant Nos. 62376110, 61977038], and Thousand Talents Plan of Jiangxi Province [Grant No. jxsq2019201124], and the Jiangxi Provincial Natural Science Foundation for Distinguished Young Scholars (20224ACB212004). The work of W. Xiao was supported in part by Jiangxi Provincial Educational Science Foundation during the"13th Five-Year Plan" (20YB080), and the Humanities and Social Science Foundation of Higher Education Institutions of Jiangxi Province(YY22209).

---

[11]https://huggingface.co/hfl/cino-base-v2

[12]https://huggingface.co/bert-base-chinese

[13]https://huggingface.co/bert-base-multilingual-cased

[14]https://huggingface.co/schen/longformer-chinese-base-4096

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

| Category | Feature name | Definition |
|---|---|---|
| Lexical level | 1: Characters | Total number of characters |
| | 2: Words | Total number of words |
| | 3: Adverbs | Total number of adverbs |
| | 4: Verbs | Total number of verbs |
| | 5: Nouns | Total number of nouns |
| | 6: Adjectives | Total number of adjectives |
| | 7: Ratio of adverbs | The ratio of the number of adverbs to the total number of words |
| | 8: Ratio of verbs | The ratio of the number of verbs to the total number of words |
| | 9: Ratio of nouns | The ratio of the number of nouns to the total number of words |
| | 10: Ratio of adjectives | The ratio of the number of adjectives to the total number of words |
| | 11: Low stroke-count characters | Total number of characters with 1-7 strokes |
| | 12: Intermediate stroke-count characters (8~15 strokes) | Total number of characters with 8-15 strokes |
| | 13: High stroke–count characters (>15 strokes) | Total number of characters with more than 15 strokes |
| | 14: Ratio of Low stroke-count characters | Proportion of ow stroke-count characters |
| | 15: Ratio of Intermediate stroke-count characters (8~15 strokes) | Proportion of Intermediate stroke-count characters |
| | 16: Ratio of High stroke-count characters (>15 strokes) | Proportion of High stroke-count characters |
| | 17: Average strokes | Total number of strokes of each character divided by the number of characters |
| | 18: Two-character words | Total number of Two-character words |
| | 19: Three-character words | Total number of Three-character words |
| | 20: Four-character words | Total number of Four-character words |
| | 21: Ratio of Two-character words | The ratio of the number of Two-character words to the total number of words |
| | 22: Ratio of Three-character words | The ratio of the number of Three-character words to the total number of words |
| | 23: Ratio of Four-character words | The ratio of the number of Four-character words to the total number of words |
| | 24: Level 0 words | Total number of words not in 8,000 Chinese Words |
| | 25: Level 0 words ratio | Lvel 0 words divided by the total number of words |
| | 26-32: Level 1,2,...,7 words | Total number of words in level 1,2,...,7 respectively |
| | 33-39: Level 1,2,...,7 words ratio | Level 1,2,...,7 divided by the total number of words respectively |
| | 40: Level 1,2,3 words ratio | The ratio of level 1,2,3 words together in all words |
| | 41: Level 4,5 words ratio | The ratio of level 4,5 words together in all words |
| | 42: Level 6,7 words ratio | The ratio of level 6,7 words together in all words |
| | 43: Average level of words | Calculate the level of all words, calculate the average level |
| | 44: Mean square level of words | Calculate the level of all words, calculate the mean square level |

Table 8: Part I of Chinese Linguistic Features

| Semantic Level | 45: Content words | Total number of content words |
|---|---|---|
| | 46: Frequency of content words | Frequency of content words |
| Syntactic level | 47: Sentences | Number of sentences |
| | 48: Average sentence length | Total number of words divided by the total number of sentences |
| | 49: Maximum sentence length | The length of the longest sentence among all sentences |
| | 50: First-level sentences | The total number of sentences with a length not exceeding 15 |
| | 51: Second-level sentences | The total number of sentences with a length between 16 and 30 |
| | 52: Tertiary sentences | The total number of sentences between 31 and 45 in length |
| | 53: Fourth-level sentences | The total number of sentences between 46 and 60 in length |
| | 54: Fifth-level sentences | The total number of sentences with a length greater than 60 |
| | 55: Ratio of First-level sentences | Ratio of the total number of sentences with a length not exceeding 15 |
| | 56: Ratio of Second level-sentences | Ratio of the total number of sentences with a length between 16 and 30 |
| | 57: Ratio of Tertiary sentences | Ratio of the total number of sentences between 31 and 45 in length |
| | 58: Ratio of Fourth-level sentences | Ratio of The total number of sentences between 46 and 60 in length |
| | 59: Ratio of Fifth-level sentences | Ratio of the total number of sentences with a length greater than 60 |
| Cohesion Level | 60: Conjunctions | Total number of conjunctions |
| | 61: Preposition | Total number of preposition |
| | 62: Pronouns | Total number of pronouns |
| | 63: Auxiliary verbs | Total number of auxiliary verbs |
| | 64: Ratio of conjunctions | The ratio of the number of conjunction to the total number of words |
| | 65: Ratio of prepositions | The ratio of the number of preposition to the total number of words |
| | 66: Ratio of pronouns | The ratio of the number of pronouns to the total number of words |
| | 67: Ratio of auxiliary verbs | The ratio of the number of auxiliary to the total number of words |

Table 9: Part II of Chinese Linguistic Features