# OpenReview forum: "PromptARA: Improving Deep Representation in Hybrid Automatic Readability Assessment with Prompt and Orthogonal Projection"
_EMNLP/2023/Conference — EMNLP 2023 Findings_

### Official Review · Reviewer_otsg · 2023-08-04

**Soundness:** 3

**Excitement:**

4: Strong: This paper deepens the understanding of some phenomenon or lowers the barriers to an existing research direction.

**Missing References:**

I didn’t find any missing references.

**Paper Topic And Main Contributions:**

To explore the deep representations and exploit the linguistic representations of texts for ARA, the paper proposed a hybrid ARA model by employing the prompts to improve deep feature representations and an orthogonal projection layer to fuse both deep and linguistic features. Extensive computational experiments on  four English and two Chinese datasets showed the effectiveness of the model.

Strengths:
1.It is demonstrated how deep feature representations and explicit syntactic information contribute to the assessment of the readability level of a document;
2.Sufficient technical details, showing the transformation process;
3.Several data-sets are included with both Chinese and English;
4.Extensive computational experiments results showed the effectiveness of the framework.

Weaknesses:
1.The function of fusion by orthogonal layers is not very clear, which means the authors need to provide examples or experimental results to demonstrate that such processing indeed removes the redundant information.

2.There are some grammatical issues with this paper, usually missing determiners and the like.

**Questions For The Authors:**

Why can the redundant information of ARA be eliminated through the fusion of orthogonal layers?

**Reasons To Accept:**

It is demonstrated how deep feature representations and explicit syntactic information contribute to the assessment of the readability level of a document;

It would be beneficial for those considering how to make use of deep features and other types of features in combination with prompts.

**Reasons To Reject:**

I don't think there much risk associated with this paper.

**Reproducibility:**

4: Could mostly reproduce the results, but there may be some variation because of sample variance or minor variations in their interpretation of the protocol or method.

**Reviewer Confidence:**

4: Quite sure. I tried to check the important points carefully. It's unlikely, though conceivable, that I missed something that should affect my ratings.

**Typos Grammar Style And Presentation Improvements:**

There are some grammatical issues with this paper, usually missing determiners and the like. I would suggest having it proofread.

---

> ### Author Rebuttal · Authors · 2023-08-28
>
> (Q1): Why can the redundant information of ARA be eliminated through the fusion of orthogonal layers?
>
> A1: The redundancy between two features (generally represented by two vectors) is roughly embodied by the correlation between them. In linear algebra, the orthogonal projection is a simple way to get rid of the correlation of two vectors. Inspired by this, this paper proposes the utilization of orthogonal projection layers to eliminate the redundant information between two different-level representations.
>
> (Q2): The function of fusion by orthogonal layers is not very clear, which means the authors need to provide examples or experimental results to demonstrate that such processing indeed removes the redundant information.
>
> A2: As presented in the response to (Q1) raised by Reviewer 3, we have conducted a series of ablation studies to demonstrate the effectiveness of prompts and orthogonal projection layers during the rebuttal phase. According to these ablation results, the improvements achieved by the orthogonal projection layers are 1.02% (Cambridge), 0.09% (weebit), 0.23% (CLEAR), 0.17% (Newsela), 1.12%(CMT), 0.78% (CMER), respectively. These results clearly demonstrate that the orthogonal projection layers can indeed remove the redundant information.
>
> (Q3): There are some grammatical issues with this paper, usually missing determiners and the like.
>
> A3: Great thanks. We will carefully proofread the paper.

---

### Official Review · Reviewer_p25R · 2023-08-04

**Soundness:** 3
**Typos Grammar Style And Presentation Improvements:** Please double-check your "Reference" …

**Excitement:**

3: Ambivalent: It has merits (e.g., it reports state-of-the-art results, the idea is nice), but there are key weaknesses (e.g., it describes incremental work), and it can significantly benefit from another round of revision. However, I won't object to accepting it if my co-reviewers champion it.

**Paper Topic And Main Contributions:**

The paper proposes to use prompts and orthogonal projection layer to improve the performance of a hybrid automatic readability assessment model that combines the neural embedding features at various granularity levels and the linguistic features.

**Reasons To Accept:**

It is a nice idea to add prompts in the learning process of ARA models.  The paper presents a neural network framework to apply prompts and orthogonal projection layers.  The framework combines learning from the representations of document-level, word-level, sentence-level, and linguistic features of input text.  Experimentation has been conducted on English and Chinese datasets to show the overall improvement of performance.

**Reasons To Reject:**

1. The paper states in lines 11 through 15 that their novelty is in the use of prompts and orthogonal projection layers. However, the ablation study is insufficient to analyze and justify the effectiveness of the prompts or the orthogonal projection layer, because:

1.1 The experimentation uses 6 datasets while the ablation study uses 4 datasets.

1.2. It is unclear from the ablation study how much improvement the prompts or the orthogonal projection layers contribute to the overall performance of the model.

First, there is no experimentation on the effectiveness of a model with orthogonal projection layers and no-prompt BigBird document representation learning. There's not enough evidential support that adding prompts will effectively improve various "no-prompt" model performance.

Second, there is no experimentation on implementing the orthogonal projection layers with no-prompt BigBird document representation learning. So we can not draw conclusions on the effectiveness of adding prompts in the implementation of the orthogonal projection layers.

Third, there is no experimentation on how each individual orthogonal projection layer help improve a prompt-enhanced BigBird model.

Since the paper states that "prompts" and "orthogonal projection" are their novelty, there needs systematic evidential support to prove the effectiveness of "prompts" and/or "orthogonal projection".

2. There is insufficient description about what kind of prompts they use, how they use them, and why these designs or choices.  The caption of Table 1 provides "Some" examples. There is no discussion how they add the "default" and non-default prompt to the documents.  Since "prompts" is the paper's major contribution, the paper needs to provide detailed discussion on their prompt engineering process.

**Reproducibility:**

3: Could reproduce the results with some difficulty. The settings of parameters are underspecified or subjectively determined; the training/evaluation data are not widely available.

**Reviewer Confidence:**

4: Quite sure. I tried to check the important points carefully. It's unlikely, though conceivable, that I missed something that should affect my ratings.

---

> ### Author Rebuttal · Authors · 2023-08-28
>
> (Q1): The ablation study is insufficient to analyze and justify the effectiveness of the prompts or the orthogonal projection layer.
>
> A1: Thanks a lot. As suggested, we have implemented a series of ablation studies over all six corpora during the rebuttal phase. In particular, we considered the following models: 1) the proposed model without prompts (denoted as w/o prompt for short), 2) the proposed model without orthogonal projection layers (denoted as w/o orth-proj for short), i.e., replacing the orthogonal projection layers with the simple concatenation, and 3) the proposed model without both of prompts and orthogonal projection layers (denoted as w/o both). In terms of accuracy, the comparison results together with the proposed model PromptARA are presented as follows: (1) Cambridge: 91.11% (PromptARA) vs. 89.14% (w/o prompt) vs. 90.45% (w/o orth-proj) vs. 88.12% (w/o both); (2) weebit: 93.12% (PromptARA) vs. 92.83% (w/o prompt) vs. 92.88% (w/o orth-proj) vs. 92.72% (w/o both); (3) CLEAR: 82.03% (PromptARA) vs. 79.28% (w/o prompt) vs. 81.62% (w/o orth-proj) vs. 79.05% (w/o both); (4) Newsela: 88.40% (PromptARA) vs. 87.53% (w/o prompt) vs. 88.12% (w/o orth-proj) vs. 87.36% (w/o both); (5) CMT: 43.96% (PromptARA) vs. 40.84% (w/o prompt) vs. 42.58% (w/o orth-proj) vs. 39.72% (w/o both); (6) CMER: 26.50% (PromptARA) vs. 24.84% (w/o prompt) vs. 25.88% (w/o orth-proj) vs. 24.06% (w/o both).
>
> It can be observed from these ablation results that there are substantial improvements on performance contributed by prompts and orthogonal layers, and the improvements contributed by prompts are generally more significant than those yielded by integrating orthogonal projection layers. These results clearly show the effectiveness of the used prompts and orthogonal projection layers.
>
> (Q2): The experimentation uses 6 datasets, while ablation studies use 4 datasets.
>
> A2: We have implemented the ablation studies over all six datasets during the rebuttal phase, as pointed out in the response to (Q1).
>
> (Q3): It is unclear from the ablation study how much improvement the prompts or the orthogonal projection layers contribute to the overall performance of the model.
>
> A3: The results of ablation studies from the response to (Q1) indicate that, in terms of accuracy, prompts yield improvements of 1.32% (Cambridge), 0.16% (weebit), 2.57% (CLEAR), 0.76% (Newsela), 2.86%(CMT), 1.82% (CMER), respectively, while the orthogonal projection layers provide improvements of 1.02% (Cambridge), 0.09% (weebit), 0.23% (CLEAR), 0.17% (Newsela), 1.12%(CMT), 0.78% (CMER), respectively. It can be observed that the improvements contributed by the prompts are more significant than those contributed by the orthogonal projection layers.
>
> (Q4): There is insufficient description about what kind of prompts they use, how they use them, and why these designs or choices.
>
> A4: Thanks for pointing out these questions.
>
> (1) What kind of prompts used: In this paper, we employ task prompts that explicitly specify the tasks the model should perform. These prompts usually take the form of questions, commands, or task statements of the target task, ensuring that the model understands the nature of the task. Given that the key goal of prompts is to explicitly inform the model what it is trying to accomplish, it’s essential for task prompts to be clear and concise, ensuring the model understands the task’s essence. Specifically, the task in this paper pertains to readability assessment. According to this design principle, we can design task prompts like "The text readability classification task is currently in progress." as presented in Table 1.
>
> (2) How to use prompts: Prompts are placed at the beginning of the text to ensure that the model explicitly understands the readability task information before reading the text. This helps the model to have a clear starting point when performing tasks.
>
> (3) Why these designs or choices: We determine these designs mainly based on the following principles. The first pertains to task-clarity—using explicit prompts ensures the model’s clear grasp of task requirements, thereby enhancing performance. The second centers on personalization—prompts should be adaptive to tasks to achieve personalized readability optimization. The third emphasizes goal-orientation—prompts should be goal-oriented, such as reducing sentence complexity or improving paragraph structure. In practice, designing and selecting prompts often requires experimentation and feedback to determine which prompts work best for a particular dataset.
>
> We will incorporate these descriptions to the paper in the future.
>
>
> (Q5): Please double-check your "Reference" list. There are many repeated references.
>
> A5: Thanks. We will double-check the reference list to avoid repeated references.

---

### Official Review · Reviewer_JPZk · 2023-08-11

**Typos Grammar Style And Presentation Improvements:** 1- In Page 7, Table 4, it’s better to…
**Soundness:** 3

**Excitement:**

3: Ambivalent: It has merits (e.g., it reports state-of-the-art results, the idea is nice), but there are key weaknesses (e.g., it describes incremental work), and it can significantly benefit from another round of revision. However, I won't object to accepting it if my co-reviewers champion it.

**Missing References:**

Missing references:
- Comprehensive readability assessment of scientific learning resources, by Arshad et al., in IEEE, 2023
https://ieeexplore.ieee.org/stamp/stamp.jsp?tp=&arnumber=10132466

- Automated assessment of subjective assignments: a hybrid approach, by Birla et al., in 2022
 https://www.sciencedirect.com/science/article/abs/pii/S0957417422006777

**Paper Topic And Main Contributions:**

Summary:
This paper extends previous ARA approach (BERT-FP-LBL [1]) by incorporating simple prompting tricks. Multiple experiments are conducted over 4 English and 2 Chinese datasets by comparing to various baselines.

[1] A unified neural network model for readability assessment with feature projection and length-balanced loss, by Li et al., in EMNLP, 2022.
https://aclanthology.org/2022.emnlp-main.504/

**Questions For The Authors:**

1- For Table 4 (Page 7), it’s better to have the finetuned results for the BERT-FP-LBL for ALL datasets. Is there any particular reason that you choose not to do that?

2- What checkpoints/pretrained weights did you use for all approaches, especially for the Chinese datasets (e.g., the BERT method in Table 5)?

**Reasons To Accept:**

-- The paper is well written.

-- The experiments are good.

**Reasons To Reject:**

-- Limited technical novelties.

-- More comparisons are needed for the proposed approach and the BERT-FP-LBL, including detailed ablation studies.

-- Instead of directly citing the results from [1] for BERT-FP-LBL, it would be more helpful if we could have the finetuned results.

**Reproducibility:**

3: Could reproduce the results with some difficulty. The settings of parameters are underspecified or subjectively determined; the training/evaluation data are not widely available.

**Reviewer Confidence:**

3: Pretty sure, but there's a chance I missed something. Although I have a good feel for this area in general, I did not carefully check the paper's details, e.g., the math, experimental design, or novelty.

---

> ### Author Rebuttal · Authors · 2023-08-28
>
> (Q1): Limited technical novelties.
>
> A1: The novelties of this paper can be summarized as follows: (1) This paper is the first work to incorporate the idea of prompt learning into the hybrid automatic readability assessment.  Leveraging this, the proposed model outperforms the state-of-the-art models over four English and two Chinese corpora. (2) The architecture of the proposed model significantly differs from those of existing models such as BERT-FP-LBL. Specifically, in the proposed model, we simultaneously use the document-, sentence-, and word-level representations to form the deep representation, while only the document-level deep representation is used in BERT-FP-LBL. The architecture of the linguistic representation module in the proposed model is also very different to BERT-FP-LBL. Notably, our model employs layer normalization to reduce the impact of different feature variations.
>
> Moreover, we have conducted experiments on the Cambridge and weebit corpora to compare the performance of BERT-FP-LBL and the proposed model without prompts (called w/o prompt for short) in the rebuttal phase. In terms of accuracy, the specific comparison results are presented as follows: (1) Cambridge: 87.78% (BERT-FP-LBL) vs. 89.14% (w/o prompt); (2) weebit: 92.70% (BERT-FP-LBL) vs. 92.83% (w/o prompt). It can be observed that the proposed model without prompts also outperforms BERT-FP-LBL. Thus, the proposed model should not be regarded as an extension of BERT-FP-LBL though both models use the hybrid representations and orthogonal projection layers for fusion.
>
> Based on the above discussions, the novelties of this paper should be deemed sufficient.
>
> (Q2): More comparisons between the proposed approach and BERT-FP-LBL are needed, including detailed ablation studies.
>
> A2: Great suggestions. In Table 4, we only present the results of BERT-FP-LBL (reported in the literature) over two English corpora since we cannot access their reproducible codes. As suggested, we have conducted some detailed ablation studies comparing the proposed approach with BERT-FP-LBL in the rebuttal phase. As mentioned in the response to (Q1), we compared the performance of the proposed method without prompts and BERT-FP-LBL. Experimental results show that the proposed method without prompts still outperforms BERT-FP-LBL over Cambridge and weebit. We would like to do more comparisons if we can access their reproducible codes in the future.
>
> (Q3): It would be more helpful if we could have the finetuned results of BERT-FP-LBL.
>
> A3: Totally agree with this. We would like to yield the finetuned results of BERT-FP-LBL if we can access their reproducible codes in the future.
>
> (Q4): For Table 4 (Page 7), it is better to have the fine-tuned results for the BERT-FP-LBL for all datasets. Is there any particular reason that you choose not to do that?
>
> A4: The results of BERT-FP-LBL presented in Table 4 were directly taken from the results reported in the literature (Li et al., 2022) since we cannot access their reproducible codes.
>
> We would like to yield the fine-tuned results for all datasets if we can access their reproducible codes in the future.
>
> (Q5): What checkpoints/pretrained weights did you use for all approaches, especially for the Chinese datasets (e.g., the BERT method in Table 5)?
>
> A5: For all approaches, we used the pre-trained (Chinese) models from the tool box huggingface.
>
> (Q6): Missing references.
>
> A6: Thanks. We will cite these references in the revised version.
>
> (Q7): It is better to add the “recall” as an evaluation metric in Table 4.
>
> A7: Thanks. We will include the results in terms of recall in the future.

---

### Official Review · Reviewer_b6jz · 2023-08-11

**Soundness:** 4

**Excitement:**

3: Ambivalent: It has merits (e.g., it reports state-of-the-art results, the idea is nice), but there are key weaknesses (e.g., it describes incremental work), and it can significantly benefit from another round of revision. However, I won't object to accepting it if my co-reviewers champion it.

**Missing References:**

References
1. Yang, Z., Xu, Z., Cui, Y., Wang, B., Lin, M., Wu, D. and Chen, Z., 2022, October. CINO: A Chinese Minority Pre-trained Language Model. In Proceedings of the 29th International Conference on Computational Linguistics (pp. 3937-3949).
2. Cui, Y., Che, W., Liu, T., Qin, B., Wang, S. and Hu, G., 2020, November. Revisiting Pre-Trained Models for Chinese Natural Language Processing. In Findings of the Association for Computational Linguistics: EMNLP 2020 (pp. 657-668).
3. Cino Huggingface Checkpoints https://huggingface.co/hfl/cino-large
4. https://huggingface.co/bert-base-multilingual-cased
4. PHollenstein, N., Pirovano, F., Zhang, C., Jäger, L.A. and Beinborn, L., 2021, June. Multilingual Language Models Predict Human Reading Behavior. In Proceedings of the 2021 Conference of the North American Chapter of the Association for Computational Linguistics: Human Language Technologies (pp. 106-123). Association for Computational Linguistics.
5. Chakraborty, S., Nayeem, M.T. and Ahmad, W.U., 2021, May. Simple or complex? learning to predict the readability of Bengali texts. In Proceedings of the AAAI Conference on Artificial Intelligence (Vol. 35, No. 14, pp. 12621-12629).
6. Jiang, T., Jiao, J., Huang, S., Zhang, Z., Wang, D., Zhuang, F., Wei, F., Huang, H., Deng, D. and Zhang, Q., 2022, December. PromptBERT: Improving BERT Sentence Embeddings with Prompts. In Proceedings of the 2022 Conference on Empirical Methods in Natural Language Processing (pp. 8826-8837).

**Paper Topic And Main Contributions:**

Readability assessment is the key step for many NLP tasks, including text simplification and educational applications.
Traditional feature-based readability formulas don't capture deep semantic meaning. Deep learning-based approaches capture semantic meaning but require a large dataset to train and finetune.

This paper proposes automatic readability assessment with prompt-based techniques in a large language model, BigBERT to address the two abovementioned challenges.

Toward that end, the authors combined prompt-based and traditional linguistic features in their experiment design. The authors conducted experiments with both English and Chinese language datasets.
Their empirical results and ablation studies demonstrate superior performance compared to baselines.


**Reasons To Accept:**

-The proposed method uses prompt-based readability assessment which overcomes the need of a large dataset for finetuning models
- Authors evaluate the proposed approach in English and Chinese language



**Reasons To Reject:**

First, I would recommend authors to compare their proposed approach with readability formula-based methods, similar to Chakraborty et al. in [6].
Second, to demonstrate the utility of the prompt-based approach, the authors can compare BERT-with prompts [7] and their proposed method: BigBERT-with prompts. Since BigBERT is a larger pre-trained model, the performance improvement over BERt is expected.


Third, for the Chinese Readability, the experiment setting is not correct. The authors can utilize the existing pre-trained Chinese models [1, 2], checkpoints for Chinese [3], and multilingual BERT [4] models.
Specifically, the results on the English dataset in Table 4 are better than the Chinese dataset in Table 5 by a large margin.

A good example of a multilingual experiment setting is Hollenstein et al. [5].


**Reproducibility:**

5: Could easily reproduce the results.

**Reviewer Confidence:**

5: Positive that my evaluation is correct. I read the paper very carefully and I am very familiar with related work.

---

> ### Author Rebuttal · Authors · 2023-08-28
>
> (Q1): Compare proposed method with readability formula-based methods.
>
> A1: Great suggestion. According to Chakraborty et al. (2021), we considered three representative readability formula-based methods, i.e., Automated Readability Index (ARI), Flesch Reading Ease (FE) and Flesch–Kincaid (FK), and compared them with the proposed method over four English corpora during the rebuttal phase. In terms of accuracy, the comparison results are presented as follows: (1) Cambridge: 16.67% (ARI) vs. 34.44% (FE) vs. 20.00% (FK) vs. 91.91% (our); (2) weebit: 19.58% (ARI) vs. 19.67% (FE) vs. 20.00% (FK) vs. 93.12% (our); (3) CLEAR: 9.35% (ARI) vs. 15.43% (FE) vs. 14.16% (FK) vs. 82.03% (our); (4) Newsela: 8.87% (ARI) vs. 7.65% (FE) vs. 8.54% (FK) vs. 88.40% (our). It can be observed that these readability formula-based methods perform worse than both the proposed method and other deep learning-based approaches, as shown in Table 4. Similar claims can be also concluded in terms of other evaluation metrics. We will incorporate these comparison results if the paper is accepted.
>
> (Q2): Compare BERT-with prompts and the proposed method.
>
> A2: We have compared BERT-with prompts (abbreviated as BERTwPrompt) with the proposed method over corpora concerned in the rebuttal phase. For better comparison, we also took the performance of BERT into consideration. In terms of accuracy, the comparison results are presented as follows: (1) Cambridge: 75.56% (BERT) vs. 78.12% (BERTwPrompt) Vs. 91.11% (our); (2) weebit: 91.53% (BERT) vs. 92.38% (BERTwPrompt) Vs. 93.12% (our); (3) CLEAR: 76.74% (BERT) vs. 79.49% (BERTwPrompt) Vs. 82.03% (our); (4) Newsela: 77.12% (BERT) vs. 84.23% (BERTwPrompt) Vs. 88.40% (our); (5) CMT: 38.46% (BERT) vs. 41.23% (BERTwPrompt) Vs. 43.96% (our); (6) CMER: 22.30% (BERT) vs. 24.14% (BERTwPrompt) Vs. 26.50% (our). It can be observed that the performance of the proposed method using the larger pre-trained BigBird model is better than that of BERT with prompts, which is also much better than that of BERT. These results highlight the efficacy of using a larger pre-trained model and the introduced prompts for improved readability assessment. Similar claims can be also concluded in terms of other evaluation metrics. If accepted, we will include these comparison results in the paper.
>
> (Q3): For the Chinese readability, the authors can utilize existing pre-trained Chinese models to improve the performance.
>
> A3: For the Chinese readability, the BigBird used in experiments is the pre-trained Chinese version of BigBird from https://huggingface.co/Lowin/chinese-bigbird-wwm-base-4096.
> As suggested, we have tried some other existing pre-trained Chinese models such as CINO and multilingual BERT (called multi-BERT for short) during the rebuttal phase. In terms of accuracy, the comparison results are presented as follows: (1) CMT: 39.19% (CINO) vs. 38.64% (Multi-BERT) vs. 43.96% (BigBird); (2) CMER: 24.37% (CINO) vs. 25.44% (multi-BERT) vs. 26.50% (BigBird). It can be observed that replacing BigBird with CINO and multilingual BERT cannot further improve the performance. We will try more pre-trained Chinese models to improve the performance of the proposed approach for the Chinese readability assessment in the future.
>
> (Q4): Some missing references.
>
> A4: Thanks. We will cite these references in the paper.

---

### Meta-Review · Area_Chair_MzDU · 2023-09-16

**Recommendation:** 4

**Metareview:**

The authors present a method for readability assessment. The key concept involves using proper prompting to induce features from a large model. The features are then combined with other features such as linguistic feature. They are combined via orthogonal projection layers. The authors observed improvements over state-of-the-art methods on four corpora.

All reviewers agree that the paper is clearly written with results seem to be easily reproducible. The evaluation on the four corpora is solid.
There is certain ambivalence of excitement from the reviewers.

Given that many NLP tasks have to deal with data scarcity challenge as the readability assessment problem, the presented approach of using prompting to generate feature from LLMs is interesting and can lead to further explorations.

---

### Decision · Program_Chairs · 2023-10-07

**Decision:**

Accept-Findings

**Comment:**

The authors present a method for readability assessment. The key concept involves using proper prompting to induce features from a large model. The features are then combined with other features such as linguistic feature. They are combined via orthogonal projection layers. The authors observed improvements over state-of-the-art methods on four corpora.

All reviewers agree that the paper is clearly written with results seem to be easily reproducible. The evaluation on the four corpora is solid.
There is certain ambivalence of excitement from the reviewers.

Given that many NLP tasks have to deal with data scarcity challenge as the readability assessment problem, the presented approach of using prompting to generate feature from LLMs is interesting and can lead to further explorations.